# Translating cognitive models into neural and statistical descriptions of real-world multi-agent foraging behavior

## Abstract

Foraging is a multi-agent social behavior that has been studied from many perspectives, including cognitive science, neuroscience, and statistics. We start from a specific type of cognitive description – agents with internal preferences expressed as value functions – and implement it as a biologically plausible neural network. We also present an equivalent statistical model where statistical predictors correspond to components of the value function. We use the neural network to simulate foraging agents in various environmental conditions and use the statistical model to discover which features in the environment best predict the agent's behavior. Our intended primary application is the study of multi-species groups of birds foraging in real-world environments. To test the viability of the statistical approach, we simulate bird agents with different preferences, and use Bayesian inference to recover what each type of agent values. In the multi-agent context, we investigate how communication of information about reward location affects group foraging behavior. We also test our modeling technique on a previously published locust foraging dataset (Günzel et al., 2023). After evaluating the effectiveness of our method on both synthetic and previously published data, we analyze new multi-agent foraging bird data we captured through high-resolution video recordings. Our method distinguishes between proximity preferences of ducks and sparrows within foraging groups. This analysis framework provides a principled, interpretable, and parametric approach for reasoning about how birds' preferences relate to their decisions about where to move in a complex multi-agent environment.

## 1 Introduction

### 1.1 Multi-species foraging behavior of small birds in winter

Small birds that spend winter in cold climates face extreme survival pressures and congregate in multi-species flocks that help them survive. In order to stay warm overnight, birds burn through about $10\%$ of their body weight, which they must regain the next day through foraging for food (Chaplin, 1974), all while avoiding predators. There is a rich literature of experimental and theoretical work on the energy management strategies that individual birds employ to survive the winter (Pravosudov & Grubb Jr, 1997; Brodin, 2007), including remembering the locations of hidden food caches, which relies on the hippocampus (Krushinskaya, 1966), a brain region also important for memory in humans (Scoville & Milner, 1957). Birds that are part of a group eat more and spend less time scanning for predators than birds that are separated from a group (Sullivan, 1984a), and they listen to other birds to decide when to forage (Sullivan, 1984b). We are interested in understanding the winter foraging behavior of multi-agent groups of birds, from a quantitative cognitive neuroscience perspective. Understanding this type of complex real-world cognitive behavior is an important component of uncovering principles of how the brain and mind work (Gao & Ganguli, 2015; Krakauer et al., 2017; Mobbs et al., 2018; Hall-McMaster & Luyckx, 2019; Miller et al., 2022; Dennis et al., 2021; Niv, 2021; Pravosudov, 2022).

Group foraging in birds involves communication of information across the group, and depends a lot on specific environmental conditions. The two primary functions of communication in wintering

bird flocks are finding food and avoiding predators. Birds are able to understand alarm and 'all clear' calls across different species (Hidalgo, 2021). Communication about food depends a lot on the environment. For example, chickadees that live in harsh high-altitude environments rely less on social information than chickadees at lower altitudes (Heinen et al., 2021), and make use of different cognitive strategies (Benedict et al., 2021). In general, social dynamics of multi-species flocks change with habitat and environmental conditions (Course, 2021; Richardson et al., 2022).

Within a multi-species flock of birds, different species may play different roles (Goodale et al., 2020). Some species lead, while 'satellite species' follow (Dolby & Grubb Jr, 1998). Similar species often form flocks together, but competitive factors sometimes discourage similar species from congregating (Sridhar et al., 2012; Marshall et al., 2012; Jones et al., 2020). In addition, some birds may be 'scroungers', benefiting from the group, but contributing little (Barnard & Sibly, 1981; Giraldeau & Dubois, 2008).

When we go from the single-agent to the multi-agent perspective, we can observe and quantify how a group of individuals behave as a whole. Learning and memory abilities of individual birds contribute to the behavior of the whole group (Falcón-Cortés et al., 2019; Nauta et al., 2020). Interestingly, both groups and individual foragers appear to reduce complex decisions about where to forage to a series of binary decisions (Sridhar et al., 2021). In order to predict the locations of foraging birds, it can be useful to include both individual-level memory of food locations, as well as a tendency to flock in groups (Kułakowska et al., 2014).

## 1.2 STATISTICAL, COGNITIVE, AND NEURAL DESCRIPTIONS OF MULTI-AGENT FORAGING

Statistical models of groups of birds have successfully captured certain behaviors using a minimal set of rules, though not foraging behaviors, to our knowledge. Three simple automaton rules are sufficient to generate surprisingly realistic-looking flocks of birds (Reynolds, 1987). Flight patterns of murmurations of starlings are well described by local neighbor interactions (Bialek et al., 2012). Some statistical models of collective animal behavior, based off of GLMs, allow for the possibility of switching between different hidden states (Bod'Ová et al., 2018; Coen et al., 2014). There has also been a long-standing effort to quantify the distribution of step sizes when animals forage, since there are statistical challenges in assessing whether or not a distribution is heavy-tailed (Edwards et al., 2007). We borrowed several aspects of these statistical models to describe multi-agent bird foraging, but we also developed Bayesian methods more appropriate for the contexts in which birds' behavior depends both on the collective behavior, and on the location of food in the environment.

In a cognitive description, foraging decisions depend on what agents find rewarding, and what they believe about the world. This has been formalized in reinforcement learning (RL) agents, which learn an internal representation of which states are rewarding, and which states are adjacent to each other. These agents make rational decisions, in the sense that they maximize the expected future value of their actions. Reinforcement learning agents perform well in a variety of foraging-related tasks (Mnih et al., 2013; Constantino & Daw, 2015; Wispinski et al., 2022).

The concept of value can be very useful for explaining foraging behavior. It offers precise calculations of what decisions are optimal in different environmental conditions (Kilpatrick et al., 2020). Furthermore, it is possible to define additional quantities that agents may value in a multi-agent setting, such as social information (Karpas et al., 2017), and to account for goals that change dynamically (Kaelbling, 1993; Todorov, 2009; Piray & Daw, 2021).

In the multi-agent foraging context, information can flow between agents. Different types of information flow generate measurably different foraging behaviors (Bidari et al., 2022). In humans, there is evidence that social network structure shapes collective cognition (Momennejad, 2022).

Social cognition has been formulated as the ability to do inverse reinforcement learning, that is, to infer another agent's internal representations from its behavior (Jara-Ettinger, 2019; Berke & Jara-Ettinger, 2021). The difficulty of inverse RL depends on the complexity of the RL task, as well as prior knowledge (Arora & Doshi, 2021). Inverse RL is achievable within the context of neural network driven foraging behaviors (Wu et al., 2020). One might imagine that different species of birds are able to infer different amounts of information from other birds. It can be possible to infer information even from the behavior of non-optimal agents (Evans et al., 2016; Zhi-Xuan et al., 2020), which is relevant in groups of birds that have different cognitive capacities and memory abilities.

Most of our understanding of the neuroscience of foraging is restricted to single agents, but there is evidence that the brain may incorporate information from other agents. At the individual agent level, there is compelling evidence across different species to suggest which neural circuits compute which information. In nematodes, there are end-to-end circuit descriptions of how the nervous system makes foraging decisions (López-Cruz et al., 2019), and basic principles from this understanding shed light on representations in the primate brain, specifically the value of possible decisions (Calhoun & Hayden, 2015). There is also evidence that the hippocampus computes the successor representation (Stachenfeld et al., 2017), an efficient representation that is useful for computing value. Interestingly, the hippocampus also generates precise representations of the locations of other agents (Omer et al., 2018). Finally, there is evidence that certain brain circuits control the decision-making part of foraging (Barack & Platt, 2017).

## 1.3 OVERVIEW

We argue that abstract cognitive descriptions of multi-agent foraging behavior can be mapped to a neural network implementation and to a statistical model. In the cognitive description, each state is assigned a value, which may be computed from a variety of features, such as the locations of food or information about other agents. We identify which components of the neural network and of the statistical model correspond to features that the agent values. Multi-agent foraging behavior varies widely across different species and environments, as discussed above. The combined cognitive/neural/statistical framework provides a family of possible descriptions of foraging behavior that can be adapted to capture different species and environments, since it is possible to add and combine many different features that may govern behavior.

The statistical perspective allows us to infer which of the proposed features best explains the behavior of a particular type of bird in a particular environment. To illustrate, we simulated different types of multi-agent groups (random-walkers, followers, and hungry birds), and found that it was possible to distinguish what different birds value by performing statistical inference that uses their simulated foraging trajectories as data.

In the multi-agent context, communication of information between agents is an important feature of group-level behavior, as is the impact of different environmental conditions. We investigated whether the benefit of communicating information would be different for different environments, using multiple simulations with a range of communication-related hyper-parameters. In environments where food was highly clustered, it took longer for birds to find food, but in all environments, using information communicated from other birds improves foraging success. We also test this communication analysis on previously published locust data (Günzel et al., 2023).

Finally, we recorded video data of multi-species multi-agent foraging birds in an outdoor winter environment. This data contains high-resolution information about birds' foraging trajectories. From these trajectories, we were are able to estimate coefficients that represent how much different species of birds value being at different proximity to other birds. These proof-of-concept analyses demonstrate that, with additional data, this analysis framework can be used to compare decision-making parameters that govern foraging behavior across different species and environments.

*The simulations, data, and analyses associated with this paper are in a github repository that will be linked after the anonymous review process. This repository is designed with modular functionality to allow the application of these analyses to new multi-agent datasets.*

## 2 RESULTS

### 2.1 TRANSLATING BETWEEN COGNITIVE, NEURAL, AND STATISTICAL DESCRIPTIONS OF FORAGING

We set out to translate an abstract cognitive description of foraging behavior into concrete empirically testable predictions about neural activity and statistics of movements. A cognitive policy can be expressed as a function that takes in a state $S$ and returns a prediction of what action $A$ the bird will do in that state. A combination of different factors may explain why an animal took a particular action. Neural and statistical descriptions of foraging also predict actions based on states, but each type of description can appear to have quite a different language and notation, so it can be difficult to

translate between them. Suppose a bird's action is influenced by the location of the food. A cognitive description might state that the bird values food. A neural description might state that hippocampal representations activate downstream neurons in a way that drives the bird to navigate toward food locations. A statistical description might state that food location is a good predictor of where the bird will move. Starting from a certain formulation of the cognitive model in terms of reinforcement learning agents, we will translate this notation into neuroscience and statistical formulations.

### 2.1.1 NOTATION

Cognitive descriptions have been formalized in reinforcement learning (RL) models. An RL agent learns an expected reward $r(S)$ for each state, as well as a 'world model' for predicting future states $S_{t+1} = T(A_t, S_t)$, where $T$ is a state transition function. The agent computes the expected future value of each possible state $V(S_{t'}) = \sum_{t=0}^{\infty} \gamma^t r(S_{t'+t})$, where $\gamma$ is a temporal discounting rate. At each step, the agent makes a rational decision to act in a way that maximizes expected future value. Specifically, at state $S$, the rational policy is to choose action $A^{cog} = \arg\max_A(V(T(A, S)))$.

A neural description passes $S$ through a neural network model of different brain areas, from perceptual areas to intermediate areas to motor areas, with the activity of the output units determining $A^{neuro}$. Specifically, $A^{neuro} = N_4(N_3(N_2(N_1(S))))$, where $N_i$ are neural network layers representing sequentially connected brain areas ranging from perceptual to motor.

A statistical description employs predictors to probabilistically predict the next action. The predictors could take arbitrary functional form, and in practice are related to observable variables, such as the location of other birds in the environment or the location of food. A statistical model of this sort might be formulated as a specification of $P(A^{stat} = A|S) = h(c_1 f_1(A, S) + c_2 f_2(A, S) + \ldots + c_n f_n(A, S))$, where $A^{stat}$ is the position where the bird will move next, $h$ is a monotonic non-linear function, and $f_i$ are predictors with coefficients $c_i$.

With this notation in mind, the goal of the next few sections is to specify biologically plausible neural networks for which $A^{neuro} = A^{cog}$, and define statistical models for which the most probable value of $A^{stat}$ is $A^{cog}$. Why is it useful to "translate" a cognitive description into a neural or statistical description? Cognitive variables are more abstract than measurable quantities such as the firing rates of neurons, or the statistics of animal movements. Once we translate abstract cognitive variables into measurable quantities, we can use data from real-world multi-agent foraging behavior to inform cognitive descriptions, which are more interpretable.

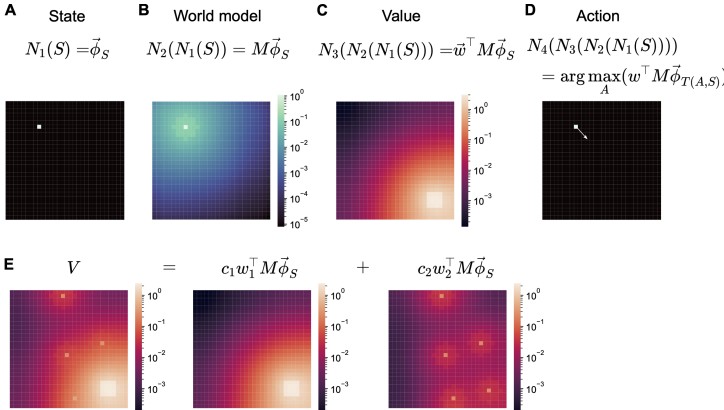

Figure 1: **Neural network implementation of a RL agent in a grid world environment.** Neural population vectors are re-shaped to match the 2D environment. **A)** State representation, by a one-hot vector of neural activity. Heatmap shows activity across the population of neurons in layer $N_1$ at state $S$. **B)** World model, in the form of a successor representation. Heatmap shows activity across the population of neurons in layer $N_2$ at state $S$. **C)** Value computation. Heatmap shows activity in $N_3$ as a function of the state. **D)** Action selection. Layer $N_4$ samples possible actions, and selects the action that maximizes value. **E)** Decomposition of value computation into multiple factors. This function is also accessible within a simple neural network architecture.

### 2.1.2 A BIOLOGICALLY PLAUSIBLE NEURAL NETWORK THAT IMPLEMENTS AN RL POLICY

A common way that RL agents compute the value function is using the successor representation, a predictive map that represents the temporally discounted expected occupancy of future states, from any starting state. In a world with $N$ states, the successor representation is an $N$ x $N$ matrix, defined as $M = \sum_{t=0}^{\infty}(\gamma J)^t = (I - \gamma J)^{-1}$, where $J$ is the transition matrix between states and $I$ is the identity matrix. In the simplest formulations, the inputs are one-hot vectors, so we construct $N_1(S)$ to be a sparsifying network that returns a population vector $\vec{\phi}_S$ that is mostly zeros, with a one at the bin corresponding to state $S$. A weight vector $\vec{w}$ stores the expected reward of each state, such that $\vec{w}^\top \vec{\phi}_S = r(S)$. The value can be computed using the successor representation in the following way $V(S) = \vec{w}^\top M \vec{\phi}_S$. This can be computed using simple linear networks $N_2 = M$, and $N_3 = \vec{w}^\top$. Interestingly, neural representations in the hippocampus, an intermediate area between perceptual and motor areas, appear to match the successor representation (Stachenfeld et al., 2017). Several mechanisms have been proposed for how this representation is computed and learned within a biologically plausible neural network (Fang et al., 2023; Bono et al., 2023; George et al., 2023). It is also possible to learn a vector of reward-related output weights using Hebbian plasticity (Zhang et al., 2021). We are almost there, having constructed a biologically plausible neural network that computes value $N_3(N_2(N_1(S))) = V(S)$. What remains is using value to inform action in a biologically plausible way.

When animals decide what action to take, they sample their local environment and evaluate their options. This process can be interpreted as an agent computing $\arg\max$. An agent at state $S$ perceives not only the current state $\vec{\phi}_S = N_1(S)$, but also adjacent states, $N_1(T(A, S))$, by actively looking or sniffing in the direction of possible actions. Action-selection circuits in the brain are thought to use winner-take-all network dynamics (Meegan, 1999) to compute the $\arg\max$, so we define $N_4 = \arg\max_A N_3(N_2(N_1(T(A, S))))$. This gives us $A^{neuro} = \arg\max_A(V(T(A, S))) = A^{cog}$.

This gives us an end-to-end simple neural implementation of an RL agent. Using this implementation, we can generate neural representations in different areas of the brain (Figure 1).

### 2.1.3 A STATISTICAL INTERPRETATION OF AN RL POLICY

We found that it is also possible to relate descriptions of RL foraging agents to certain statistical descriptions. The use of predictors in such descriptions may at first seem antithetical to RL methods, where predictions are based on a single quantity—'reward'. However, RL descriptions often decompose rewards into a collection of different factors. In the case of multi-agent groups of birds, these factors could include not only food locations, but other quantities such as the caloric cost of moving, the relative exposure to predators of different locations, and the potential benefit or cost of being near other birds. This decomposition into factors makes RL descriptions appear more similar to the statistical descriptions (Figure 1). For further detail of the statistical formulation, and relation to cognitive and neural descriptions, see Appendix A.

## 2.2 SIMULATIONS OF MULTI-AGENT FORAGING BEHAVIOR

### 2.2.1 STATISTICAL INFERENCE TO ESTIMATE WHAT FACTORS BIRDS VALUE

We first simulate three different types of foraging groups: hungry, follower and random birds. These groups correspond to three types of things birds might value – food, proximity to other birds, and nothing. Our goal was to test whether it is possible to infer what the agents value from observing their movements and the rewards locations. We fit a Bayesian model in which possible components of a value function (food and proximity to other birds) are expressed as statistical predictors. First, we compute the predictors for a range of locations in space and time. Example proximity scores and food traces are shown in Figure 2. These are then used to predict where each bird will go next, by fitting a Bayesian model that predicts, for each frame, how close locations are to the birds' locations at the next step. The mean prediction is modeled linearly as a function of a baseline and the predictors, with the addition that, to handle heteroskedasticity, raw standard deviation is modeled linearly in an analogous manner, and then passed through a softmax function. For further detail on the construction of the derived predictors, see Appendix B. The coefficients estimated using this model allow us to to clearly disambiguate random, hungry, and follower birds (Figure 2).

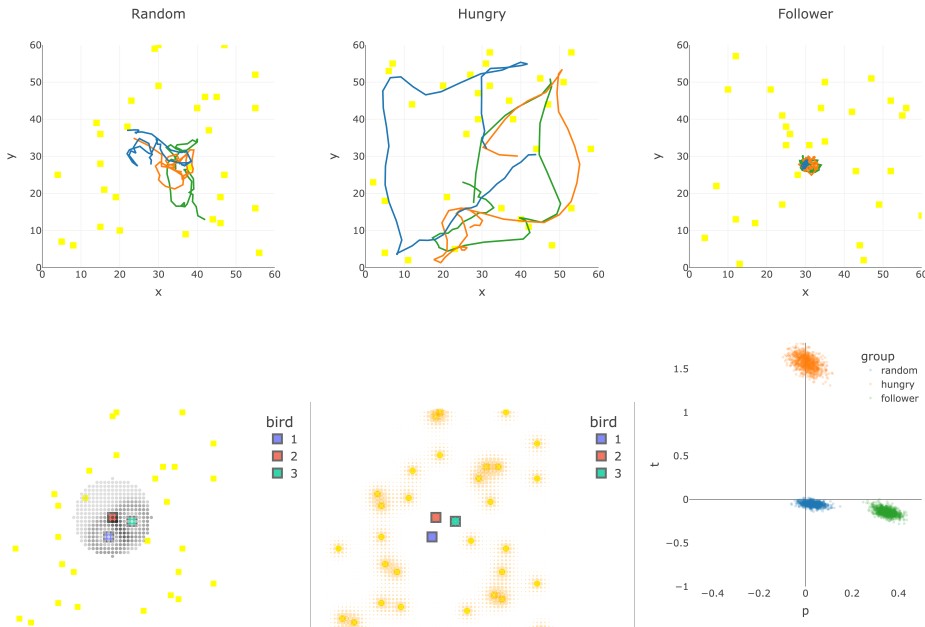

Figure 2: **Inferring the values of random, hungry, and follower birds** (Top) Example trajectories for random, hungry, and follower birds. (Bottom left) Proximity scores (gray) for points visible to bird 2 at a single time frame. Yellow indicates food locations. (Bottom center) Food traces for the same frame. (Bottom right) Coefficient values for proximity (p) and trace (t), sampled from the posterior estimated using SVI on synthetic datasets of random, hungry, and follower birds.

### 2.2.2 MODELING THE EFFECT OF INFORMATION-SHARING ON GROUP FORAGING SUCCESS, ACROSS DIFFERENT ENVIRONMENTS

Next, we investigate the effect of information-sharing on foraging success under different environmental conditions. In real-world environments, animals may use not only their own sensory information but also communication with other animals to inform their decisions, for example visual observations, or listening to other birds' calls. We simulated grid world environments with food patches of varying degrees of spatial clustering, controlling for the total amount of food in the environment. In each environment, there were 16 total food items, distributed randomly in patches (1x1, 2x2, or 4x4).

We parameterized the extent to which agents share information about food locations. Agents follow a policy $A = \arg\max_A(V(T(A, S)))$, where the expected reward $r(S)$ depends on the location of visible food in the environment. Each bird is only able to see food within a radius of 5 steps. Birds also update their expected reward vector $\vec{w}$ at the locations of other birds that are at food locations. In the real world, this could be achieved by observing other birds and listening to their calls. The weighting of social information (reward locations communicated by other birds), compared to individual information, is given by the communication parameter, which ranges from 0 (no communication) to 1 (full reliance on social information). Birds that communicate appear to navigate more directly to food locations than birds that search independently (Figure 3, left). We analyzed these simulated birds using a similar model as in the Random/Hungry/Follower bird case, but with an additional communication term, which updates when other birds arrive at food locations (Appendix C).

The model correctly assigns near-zero weight (c) to the communication term for simulations with non-zero values for the communication parameter, but not for simulations with zero values of the communication parameter (Figure 3, center). Note that the weight of the proximity score (p) is also non-zero for communicator birds.

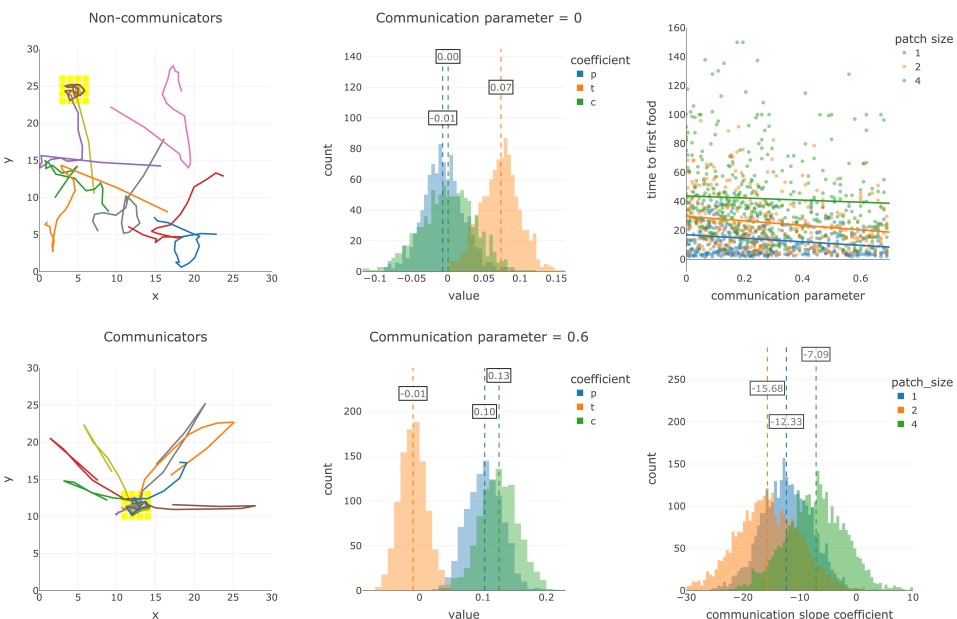

Figure 3: **Modeling the effect of information-sharing on group foraging outcomes.** (Left) Trajectories of simulated birds that either search for food independently (top), or communicate food locations (bottom, communication parameter = 0.06). Reward locations are in yellow, with patch width 4. (Center) Posterior distributions of fit coefficients for proximity (p), food trace (t), and communication (c), for non-communicators (top), and communicators (bottom, communication parameter 0.06). Medians shown as dashed lines. (Right, top) Average time to first find food, across simulated groups of birds with different communication parameters. Colors indicate the size of the food patches in the environment. Lines indicate linear fits obtained using regularized Bayesian models (see Appendix C). (Right, bottom) Posterior distribution of slopes of the linear model, estimated using Stochastic Variational Inference (SVI). Across environments with different patch sizes, there are negative estimated slope coefficients, that is, increased communication corresponds to decreased time to first food.

In order to assess the role of communication across different environments, we simulated groups of birds with different communication parameters, across a range of environments with different degrees of food clustering. Foraging success was measured by the average time it took birds to reach their first food item. Overall, finding food took longer in environments where food was clustered into larger patches. In all environments tested, communication was negatively correlated with the time it took to reach food (Figure 3, right).

## 2.3 REAL MULTI-AGENT FORAGING DATASETS

### 2.3.1 LOCUST FORAGING IN CONTROLLED ENVIRONMENTS

Before testing this analysis framework on new real-world bird data, we test it on previously published locust data from Günzel et al. (2023). A particularly interesting claim of that paper is that locusts integrate socially derived information (observations of other locusts feeding) in their decisions of where to forage. Gunzel et al. model the locust decision mechanisms using a drift diffusion dynamical systems model. Our goal is to show that our method can replicate this general conclusion, but achieving it using a much simpler strategy.

We asked whether our cognitive RL model and Bayesian inference procedure could identify the use of social information in the same locust behavior data set. To model the locust behavior, we use the same model as we used to analyze simulated bird communication (Figure 3), with parameters set to mimic the assumptions of the paper (such as locust can see quite far, etc.), illustrating how expert knowledge can be relatively easily integrated with our framework. The posterior distributions

obtained from Bayesian inference indicate that there is indeed social information used in the locusts' decisions (Appendix D, Figure 5). This illustrates the point that while simple, our framework can still capture a relatively rich real world behavior.

### 2.3.2 NEW DATASETS TRACKING GROUPS OF BIRDS FORAGING

Studying real-world cognitive behaviors such as avian cognition is essential for understanding cognition and the brain, but it can be challenging to acquire this necessary data (Gao & Ganguli, 2015; Krakauer et al., 2017; Mobbs et al., 2018; Hall-McMaster & Luyckx, 2019; Miller et al., 2022; Dennis et al., 2021; Niv, 2021; Pravosudov, 2022). In larger birds such as cormorants, GPS trackers can continuously record foraging movements (Cook et al., 2017), but these devices weigh more than many small wintering birds, so would be impossible for them to carry. Machine vision has emerged as a revolutionary new technology for tracking animal behavior (Couzin & Heins, 2022; Naik et al., 2023). These methods are limited by the field of view of the cameras, so large-scale movements such as migration cannot be tracked, but within the field of view they track behavior at high spatiotemporal resolution, with the added benefit of being non-invasive. Foraging behavior occurs on a smaller scale than migration, so is amenable to these methods.

We acquired videos of multi-agent multi-species groups of birds performing foraging behavior in the winter. Standard RGB videos were recorded simultaneously with thermal videos (FLIR E54 camera), which can be especially effective in detecting movements of birds (Matzner et al., 2020). The RGB videos are used for identifying terrain and species, while the thermal videos are used for tracking the movement of birds (Figure 4, top left). Even in a complex natural setting with a highly varying background, the thermal videos give us low-background position information, since birds stand out as warm objects against the cold ground. The RGB videos capture terrain information about the environment that birds are foraging in, and enable the identification of different species, using automated systems (Van Horn et al., 2015). It is possible to observe trajectories of foraging birds by computing a simple maximum projection of thermal images (Figure 4, bottom left). In addition, we adapted a deep-learning-based multi-agent tracking pipeline (Pereira et al., 2022) to automatically track bird locations in these videos (Figure 4, center left).

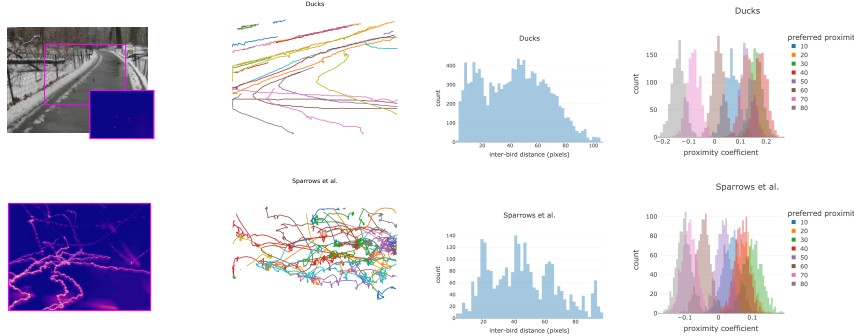

Figure 4: **Tracking of multi-species foraging behavior, using concurrent RGB and thermal imaging**. (Left, top) RGB videos are used for identifying terrain and bird species, and thermal videos are used for tracking bird movements. (Left, bottom) Maximum projection of approximately 30 seconds of thermal video data, showing the trajectories of birds. (Center-left) Trajectories of a group of ducks (Mallards), and a mixed-species group of White-Throated Sparrows and Tufted Titmice. (Center-right) Histograms of inter-bird distances for the two groups. (Right) Inferred posterior distributions of the proximity coefficient for different settings of the preferred proximity.

We wondered whether our analysis framework could recover differences in foraging preferences between different groups of birds. We chose two short video recordings, one of a group of Mallard ducks, and one of a multi-species group of small songbirds (White-Throated sparrows and Tufted Titmice). First, we observed that the different groups exhibited different distributions of inter-bird distances. We wondered whether this could be explained by different foraging preferences, specifically different preferred proximity (Figure 2 plots example proximity functions, detailed definition Appendix B). We fit variations of the model described above, with different settings for the preferred

proximity distance (Ranging from 10 pixels to 80 pixels. An important future direction will be using a multi-camera setup to convert these trajectories to 3D coordinates). In the duck data, proximity preference of 40 pixels had the strongest weighting, while 30 pixels carried the strongest weighting for the sparrow and titmouse dataset. Interestingly, both datasets showed a negative weighting for large distances (i.e., birds preferred to avoid large separations). These results were consistent with the empirical distributions of inter-bird distances, but with added explanatory power, as they are fit on individual foraging decisions, and allow for the possibility that some common inter-bird distances may not be strongly predictive of where birds will go. While more data is necessary to resolve how well these preference profiles generalize to other settings, this framework offers a promising toolkit for inferring what preferences drive behavior within real-world multi-agent behaviors.

## 3    DISCUSSION

We have presented a strategy for bridging cognitive, statistical, and neural descriptions of multi-agent foraging behavior. Starting from an abstract cognitive description of how each agent assesses what is valuable and decides what to do, we implemented each agent as a biologically plausible neural network. We simulated a group of these biologically plausible agents across different environments. From the statistical perspective, each statistical predictor corresponds to a component of the cognitive agent's value function. Using the statistical model, we could infer properties of the value functions of different agents. Finally, we collected high-resolution thermal and RGB videos of multi-agent multi-species groups of birds foraging in an outdoor setting. These results pave the way for a more integrated and comprehensive understanding of multi-agent foraging behavior by offering a family of descriptions that can be adapted to capture different species and environments.

What foraging strategies do specific groups of birds actually use? The answer to this question will differ for different species and environments. It depends on variables such as the birds' diet, the distribution of food in the environment, and the birds' cognitive capacity. For example, different species of overwintering birds have different spatial memory abilities and relative hippocampus sizes and recruitment (Hampton & Shettleworth, 1996; Hoshooley & Sherry, 2007). Within the unified framework outlined in this paper, it will be possible to combine information about relative hippocampus size with information about movement statistics and food preferences. In the future, it will also be important to relate the framework described here to evolutionary perspectives.

A key question in multi-agent foraging is what information birds communicate. Addressing this question requires understanding not only the birds' capacity for social cognition, but also under what situations it is beneficial to share certain types of information. When assessing birds' capacity for social cognition, it may be necessary to observe them in environments where social cognition is particularly beneficial.

Multi-species bird foraging, in particular, offers many interesting technical challenges. Building off of recent progress in machine vision for animal tracking (Graving et al., 2019; Pereira et al., 2022; Lauer et al., 2022; Sun et al., 2022), we will need to extend machine vision tracking methods to large 3D settings, at high enough resolution to identify particular species (Van Horn et al., 2015), or even individuals (Ferreira et al., 2020).

Observing birds' movements alone will in general not be sufficient to determine what cognitive strategies they are using. In order to properly constrain our models, we need prior information about distributions of food, movement statistics, metabolic rates, calorie consumption, and hippocampus size. We need powerful inference algorithms to capture this rich information. Two particularly promising strategies involve the ability to infer hidden states that drive behavior (Linderman et al., 2017) and the ability to specify generative models programmatically (Cusumano-Towner et al., 2017; Bingham et al., 2019; Das et al., 2023). With so many possible strategies for multi-agent foraging behavior, we need to quantify the uncertainty in our models and find automatic ways to bridge different types of models and descriptions. Our work presented in this paper is just a step in this direction.

One of the oldest functions of the embodied nervous system is exchanging information within multi-agent groups of animals, in particular about the location of food, predators, and mating compatibility. Over evolutionary time, collaborative exchange of information becomes increasingly sophisticated as ecosystems and societies expand. So, at least in principle, the key aspects of our modeling

approach are not limited to avian or insect subjects. The simulations and analyses here involved small or medium-sized groups of agents foraging in 2D grid-world environments, which could apply to a variety of multi-agent groups across the animal kingdom. The value function can be adapted to a particular animal clade by modifying the internal components based on features particular to that animal's environment and its particular neural and cognitive resources.

This framework could potentially be applied to studying foraging behaviors of groups of humans. Studies of decision-making in humans use similar neural and cognitive models as the ones we use here. For example, RL models of human planning based on the hippocampus also use successor representations (Momennejad, 2020; De Cothi et al., 2022). Models of humans performing visual information-foraging tasks may decompose the value function into different visual features each given a different attention weight (Radulescu et al., 2019).

Many real-world multi-agent behaviors, such as migratory behaviors (Hanson et al., 2022), span huge distances, sometimes at a global scale. One potential challenge lies in the sparsity of data, even if we restrict our attention to information about subjects' locations and availability of rewards. Another challenge is computational complexity, which is bound to increase with the larger spatiotemporal scale at play. Given their massive complexity and scale, these behaviors may need to be understood using more abstract descriptions, perhaps a combination of different descriptions.

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

# A APPENDIX: DERIVING CORRESPONDENCES WITH THE STATISTICAL INTERPRETATION

## A.1 RELATIONSHIP BETWEEN STATISTICAL AND COGNITIVE DESCRIPTIONS

We will express an RL policy in terms of predictors that can be derived from empirical data. As in the neural network formulation, we will focus on the successor representation formulation of an RL agent, but here we only consider the case when the agent has already learned the environment and is executing a rational policy. The expected reward of each state is stored in a vector $\vec{w} = a_1\vec{w}_1 + a_2\vec{w}_2 + \ldots + a_n\vec{w}_n$, where $a_i$ are scalar coefficients, and $\vec{w}_i$ are vectors representing different factors that contribute to the overall reward of a foraging bird, such as the quantity of food, conspecifics, or predators at each state. We can re-write the value function using the matrices and vectors defined in the successor representation as follows $V(S) = \vec{w}^\top M \vec{\phi}_S = \sum_i a_i \vec{w}_i^\top M \vec{\phi}_S$, recall that $M$ is the successor representation matrix, and $\phi_S$ is a one-hot vector representation of state $S$. Now, we are able to get that the most likely value of $A^{stat}$ (which we call $A_s$) is $A^{cog}$, when we set the statistical predictors $f_i(A, S) = w_i^\top M \vec{\phi}_{T(A,S)}$ and let $c_i = a_i$ (the first step is by the monotonicity of $h$).[1]

$$A_s = \arg\max_A h(\sum_i c_i f_i(A, S)) \tag{1}$$

$$= \arg\max_A (\sum_i c_i f_i(A, S)) \tag{2}$$

$$= \arg\max_A (\sum_i a_i w_i^\top M \vec{\phi}_{T(A,S)}) \tag{3}$$

$$= \arg\max_A (w^\top M \vec{\phi}_{T(A,S)}) \tag{4}$$

$$= \arg\max_A (V(T(A, S))) = A^{cog} \tag{5}$$

We can interpret the different statistical predictors as different components of the value function of an RL agent. For example, we can switch between describing a bird as valuing food and describing food as a good predictor of where the bird will go. This correspondence allows us to use statistical inference techniques within the context of abstract cognitive models.

## A.2 RELATIONSHIP BETWEEN STATISTICAL AND NEURAL DESCRIPTIONS

Our focus here is is how the brain may decompose value computations into different factors, similar to how statistical descriptions make use of multiple different predictors. We know now that $A^{neuro} = A^{cog} = A_s^{stat}$. In the previous section, we decomposed the expected reward into factors $\vec{w} = a_1\vec{w}_1 + a_2\vec{w}_2 + \ldots + a_n\vec{w}_n$. This could be implemented in neural network output weights, onto $n$ output neurons instead of one, with each output neuron computing a different component of value (Figure 1E). This type of output from a hippocampus-like structure was recently proposed as part of the Endotaxis model (Zhang et al., 2021). A benefit of this architecture is that the brain could flexibly learn each of the different output associations (through Hebbian plasticity), and flexibly modify

---

[1]One proviso: this identification works if the statistical model uses maximum *a posteriori* estimates with fairly wide priors; otherwise, the most likely value recommended by the statistical model might be slightly different from the one recommended by an RL model that does not incorporate whatever information that was used to shape the priors.

which output units should influence behavior, so that the animal might sometimes value food ($a_{food}$ is high), and at other times value shelter ($a_{shelter}$ is high). By bridging between the neural and statistical perspectives, we can relate this neural description of outputs from the hippocampus to statistical descriptions - neural representations in the hippocampus correspond to hidden states that can influence the coefficients of different predictors.

## B   APPENDIX: RANDOM, HUNGRY AND FOLLOWER BIRD SIMULATION AND INFERENCE METHODS

### B.1   THE THREE GROUPS

The three different types of foraging groups in our simulation are as follows:

- **Random birds.** The birds just perform random walks.
- **Follower birds.** Bird 1 does a random walk as in the previous simulation, the other birds at each step move to a location close to where Bird 1 was one step before (that is, they follow her with lag 1).
- **Hungry birds.** Birds ignore each other and pursue rewards as fast as possible (with some stochastic element, at each step randomly picking their next step from five top efficient paths available to them).

We start with simulating synthetic data that includes only locations of three or four birds (in the follower birds' case) and the positions of multiple rewards across time (rewards disappear when approached by a bird). Each simulation is on a 100x100 grid with 150 time-frames.

### B.2   DERIVED PREDICTOR SCORES

Given each dataset, for each bird, we assign three scores for each location in the environment - visibility, trace (based on distance from rewards), and proximity to other birds.

We first use an external visibility range hyper-parameter, which determines how far the birds can "see" to assign non-zero visibility scores to points in birds' vicinity. We employ a cosine decay function so that birds can better see things that are closer. Each visible location is assigned a **trace** score: 1 if it contains a reward, with exponential decay for distances farther away from rewards (length constant 6 units). To capture birds' preferences for proximity to other birds, we construct a **proximity** score function as a piece-wise function built from two sine functions and one exponential function, which assigns a negative value to being too close to another bird, but a positive value to being not too far from here (Figure 2). The proximity score is parameterized by three numbers: the x-intercept, what the optimal distance is, and an exponential decay rate for larger distances. Scores assigned to points accumulate additively across sources (e.g. being close to two rewards results in higher trace score).

### B.3   INFERENCE TASK

The prediction task is as follows: Each bird $b$ and each time frame $t$, given her range, uniquely determines the points available to $b$ at $t$, $\{p_i = \langle x_i, y_i \rangle | p_i \in \mathsf{Range}(b,t)\}$. Each such $p_i$ gets a trace score, $\mathsf{trace}(p_i)$, and a proximity score, $\mathsf{proximity}(p_i)$. At time $t+1$ the bird moves to a new position $p_{\langle b,t+1 \rangle}$. What we ideally want to be able to predict using $\mathsf{trace}(p_i)$ and $\mathsf{proximity}(p_i)$ is a transformed distance of $p_i$ from where the bird will go next, $p_{\langle b,t+1 \rangle}$, for all points $p_i$ in $\mathsf{Range}(b,t)$. More formally, as the output variable we take $\mathsf{accuracy}(\mathsf{p_i}) = - \left[ \frac{(x_i - x_{\langle b,t+1 \rangle})^2 + (y_i - y_{\langle b,t+1 \rangle})^2}{\max\left[(x_i - x_{\langle b,t+1 \rangle})^2 + (y_i - y_{\langle b,t+1 \rangle})^2\right]} \right] + 1$, so that score 1 is assigned to the point to which the bird will go next, and score 0 is assigned to the available points that are the furthest to where she will go next.

Given the scale of the grid, we fixed the hyper-parameters at fairly sensible values.[2] For each dataset the model was first fed the raw data, then used the hyper-parameters to assign trace and proximity

---

[2]Such as: rewards decay = .5, visibility range = 9, maxStepSize = 4, negative proximity score starts at 1.5, optimal proximity = 3, proximity decay = 1. Note that proximity score was not used in the simulation

scores to points in birds' ranges, which were subsequently normalized by dividing by their maximal values to ensure equal coefficient interpretability.

# C  APPENDIX: BAYESIAN MODELS

Here we provide the model code for models used in the paper. All the models other than the one described in in Listing 3 were trained on derived predictors.

## C.1  RANDOM-HUNGRY-FOLLOWERS

Listing 1: Model used to distinguish the predictive role of trace and proximity in the behavior of simulated birds: p, t, v, b are the coefficients for proximity, trace, visibility, and the intercept, ps, ts, vs, bs are analogous coefficients, but they contribute to the variance, which is not assumed to remain fixed.

```python
def model_sigmavar(proximity, trace, visibility, how_far_score):
    p = pyro.sample("p", dist.Normal(0, .8))
    t = pyro.sample("t", dist.Normal(0, .8))
    v = pyro.sample("v", dist.Normal(0, .8))
    b = pyro.sample("b", dist.Normal(.5, .7))

    ps = pyro.sample("ps", dist.Normal(0, .8))
    ts = pyro.sample("ts", dist.Normal(0, .8))
    vs = pyro.sample("vs", dist.Normal(0, .8))
    bs = pyro.sample("bs", dist.Normal(.2, .6))

    sigmaRaw = bs + ps * proximity + ts * trace + vs * visibility
    sigma = pyro.deterministic("sigma", F.softplus(sigmaRaw))
    mean = b + p * proximity + t * trace + v * visibility

    with pyro.plate("data", len(how_far_score)):
        pyro.sample("obs", dist.Normal(mean, sigma), obs=how_far_score)
```

## C.2  COMMUNICATING BIRDS

Listing 2: Model used to distinguish the predictive role of communication, trace and proximity in the behavior of simulated birds. Coefficients as above, with the addition of c and cs as communication-related coefficients.

```python
def model_sigmavar_com(proximity, trace, visibility,
            communicate, how_far_score):

    p = pyro.sample("p", dist.Normal(0, .6))
    t = pyro.sample("t", dist.Normal(0, .6))
    v = pyro.sample("v", dist.Normal(0, .6))
    c = pyro.sample("c", dist.Normal(0, .6))
    b = pyro.sample("b", dist.Normal(.5, .6))

    ps = pyro.sample("ps", dist.Normal(0, .6))
    ts = pyro.sample("ts", dist.Normal(0, .6))
    vs = pyro.sample("vs", dist.Normal(0, .6))
    cs = pyro.sample("cs", dist.Normal(0, .6))
    bs = pyro.sample("bs", dist.Normal(.2, .6))

    sigmaRaw = bs + ps * proximity + ts * trace +
                    vs * visibility + cs * communicate

    sigma = pyro.deterministic("sigma", F.softplus(sigmaRaw))
```

of follower birds. Sanity checks can be performed by running animations visualizing the predictors using the notebooks on github.

```
20      mean = b + p * proximity + t * trace +
21              v * visibility + c * communicate
22
23      with pyro.plate("data", len(how_far_score)):
24          pyro.sample("obs", dist.Normal(mean, sigma), obs=how_far_score)
```

Listing 3: A simple linear model used to evaluate the impact of communication levels on a success measure (average time to first reward).

```
1   def model(trust, time, patch):
2
3       with pyro.plate("coefs", 3):
4           base = pyro.sample("base", dist.Normal(base_m, base_s))
5           slope = pyro.sample("slope", dist.Normal(slope_m, slope_s))
6
7       sig = pyro.sample("sig", dist.LogNormal(4, 0.7))
8
9       with pyro.plate("obs", len(time)):
10          pyro.sample("time", dist.Normal(base[patch] +
11              slope[patch] * trust, sig), obs=time)
```

### C.3 Central park birds

Listing 4: A model used to evaluate the impact of proximity (with various different hyperparameter settings) together with information about distance of a location from the birds' position on how far that location was at the next step from where the birds actually went. Trained on predictors derived from the real-life data we collected.

```
1   def model_sigmavar_proximity(distance, proximity, how_far):
2       d = pyro.sample("d", dist.Normal(0, .6))
3       p = pyro.sample("p", dist.Normal(0, .6))
4       b = pyro.sample("b", dist.Normal(.5, .6))
5
6       ds = pyro.sample("ds", dist.Normal(0, .6))
7       ps = pyro.sample("ps", dist.Normal(0, .6))
8       bs = pyro.sample("bs", dist.Normal(.2, .6))
9
10      sigmaRaw = bs + ds * distance +  ps * proximity
11      sigma = pyro.deterministic("sigma", F.softplus(sigmaRaw))
12      mean = b + d * distance +  p * proximity
13
14      with pyro.plate("data", len(how_far)):
15          pyro.sample("obs", dist.Normal(mean, sigma), obs=how_far)
```

### C.4 Locust

Listing 5: A sequence of models used to evaluate the impact of proximity, trace and communication on where the locust move at the next step. This real-world dataset is highly irregular with multi-colinearity. So we dropped the linearity assumption, paritioned the dataset at regular intervals of predictor values and run a version of kernel density estimation to separately estimate means and standard deviations at the cells. We did so using three independend models, as colinearity was too prohibitive to use all the predictors in the same model. Next, we passed on the resulting mean estimates to standard weighted linear regression models, whose weights were inverses of standard deviations at given cells (the less variance in a cell, the more it contributes to the slope).

```
1
2   def discretized_p(proximity_id, how_far):
3       p = numpyro.sample("p", dist.Normal(0, 0.5).expand(
4               [len(set(proximity_id))]))
5       sigma = numpyro.sample("sigma", dist.Exponential(1))
```

```
 6          mu = p[proximity_id]
 7          numpyro.sample("how_far", dist.Normal(mu, sigma), obs=how_far)
 8
 9      def discretized_t(trace_id, how_far):
10          t = numpyro.sample("t", dist.Normal(0, 0.5).expand(
11                  [len(set(trace_id))]))
12          sigma = numpyro.sample("sigma", dist.Exponential(1))
13          mu = t[trace_id]
14          numpyro.sample("how_far", dist.Normal(mu, sigma), obs=how_far)
15
16      def discretized_c(communicate_id, how_far):
17          c = numpyro.sample("c", dist.Normal(0, 0.5).expand(
18                  [len(set(communicate_id))]))
19          sigma = numpyro.sample("sigma", dist.Exponential(1))
20          mu = c[communicate_id]
21          numpyro.sample("how_far", dist.Normal(mu, sigma), obs=how_far)
22
23      guide_p = AutoLaplaceApproximation(discretized_p)
24      guide_t = AutoLaplaceApproximation(discretized_t)
25      guide_c = AutoLaplaceApproximation(discretized_c)
26
27  #second layer, used mostly for interpretability
28
29  lr_p, lr_t, lr_c = [LinearRegression() for _ in range(3)]
30
31  X_p = np.array(summary["id_p"]).reshape(-1, 1)
32  X_t = np.array(summary["id_t"]).reshape(-1, 1)
33  X_c = np.array(summary["id_c"]).reshape(-1, 1)
34
35  lr_p.fit(X_p, summary["params_p"], sample_weight=1/summary["std_p"])
36  lr_t.fit(X_t, summary["params_t"], sample_weight=1/summary["std_t"])
37  lr_c.fit(X_c, summary["params_c"], sample_weight=1/summary["std_c"])
38
39  # resampling to gauge the uncertainty involved
40  # repeated for all coefficients
41
42  coef_samples = []
43      for _ in range(1000):
44          X_resampled, y_resampled = resample(
45              input, summary[f"params_{coef}"],
46              random_state=np.random.randint(1000)
47          )
48
49          model.fit(X_resampled, y_resampled)
50          coef_samples.append(model.coef_[0])
```

## D  APPENDIX: LOCUST DATA

The communication inference analysis in Figure 3 was performed on multi-agent locust foraging data from Günzel et al. (2023). Consistent with the original results, this analysis recovers that locusts use both individually and socially derived information to guide foraging decisions.

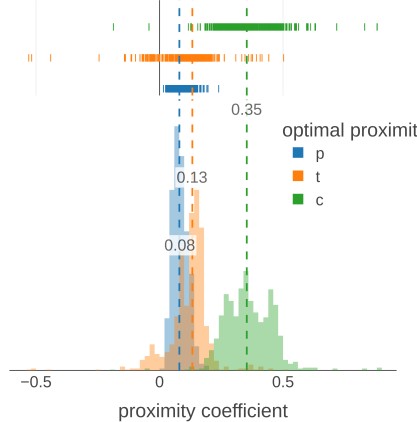

Figure 5: Posterior distributions of derived coefficients of food trace (t), proximity score (p), and communication (c) for locust data.

