# OpenReview forum: "Translating cognitive models into neural and statistical descriptions of real-world multi-agent foraging behavior"
_ICLR.cc/2024/Conference — Submitted to ICLR 2024_

### Official Review · Reviewer_3d6u · 2023-10-15

**Soundness:** 3 good
**Presentation:** 1 poor
**Contribution:** 3 good
**Rating:** 5
**Confidence:** 2

**Summary:**

The authors posit that foraging can be expressed as a value-maximizing behavior, where the agent choses the action that maximizes value, i.e. discounted sum of future rewards (for an adequate reward function and transition matrix).

They then show that this strategy can be implemented by a neural network (with some added machinery), and captured by a statistical model based on hand-crafted predictors. IIUC, the hope is that the neural model may possibly guide neurobiological investigations, while the statistical model can be fit to real-world data and provide explanatory power.

As a demonstration, they fit the statistical model to data from both simulated and real-world data, including, apparently, a completely new dataset. The model seems able to capture and differentiate underlying structure in the agent's behaviors.

**Strengths:**

- The methods seem novel.

- The resulting model(s) seems promising

- The application of the statistical model to real-world data is interesting, and seems to provide some information.

- Basically, the scientific content, as far as I managed to understand it, seems interesting and potentially valuable.

**Weaknesses:**

- The paper is clearly unpolished and, frankly, unfinished. The description is extremely confusing and incomplete (see below). It took me a lot of time to understand what the paper was trying to do. (I'm still not sure I got it). More pressingly, it seems impossible to replicate the proposed method from the inadequate description provided in the paper.

- Simply understanding what the authors try to do exactly is a challenge, because the explanation of the various parts of the method are split into multiple parts (and the Appendix), The authors don't explain what they call a "cognitive", "neural" or "statistical model" until page 4, in the Results.

- The introduction and discussion are very lengthy and philosophical, while the figures are absurdly small and the description of the methods either inadequate or just plain missing. The most basic explanation of what is actually going on is largely pushed to the Appendix. Even then, the explanation is not sufficient to reproduce the method. See Questions, below.

**Questions:**

- p.4: "f_i are predictors" - what's a "predictor"? A function? Is it linear? Monotonic?

- The successor representation is described as "Discounted expected occupancy" - but this is notoriously policy-dependent. Instead, the authors actually base theiur successor representation M on the iterated transition matrix (action-independent, IIUC). so it's really a distance function! (it might be understood as "occupancy under random policy")

- In the neural model, is M specified by hand or is it learned ? Appendix A.1 seems to indirectly imply that it is learned (whereas it is fixed for the statistical model) - but how?

- The statistical model is only described in the Appendix (and even so inadequately, see below). Before seeing the appendix, we have no idea what predictors are being used, what coefficients are, etc. Yet these are constantly referred to in the rest of the text, and indeed constitute the main target quantities in the experiments! This is clearly not workable.

- At no point is the statistical model clearly explained, with a clear description of all the predictors and coefficients and of what, exactly, is fit to the data as opposed to being hand-provided.

- The authors simulate "hungry, follower and random" birds. How? (It's briefly discussed in the Appendix, which is not referred to in the main text).

- As another example of the inadequate description, it is impossible to get a clear picture of what the proximity function (which is fundamental to the method)  actually looks like. How are the two sine waves and the exponential combined?

- In 2.2.2, where the estimated values of certain parameters are reported, it doesn't tell us what is the true value of the parameters! The results show 0 effect of "food trace" in the communicators' case, and 0 for proximity in the non-communicators. Is that true in the simulation? That is not mentioned in the text.

- What is a "maximum projection of thermal images"?

- Figure 4 shows that various values of the preferred proximity parameter give rise to different fitted estimations for the proximity coefficient. This seems to be a very cumbersome way to approximately recover the true underlying proximity-preference function, which could probably better be done with a different parametrization (E.g. a polynomial times decaying exponential, which would allow the model to fit both the peaks and troughs of the proximity function with only a few more additional parameters?)

- Generally, in the figures: "Top", "Bottom", "Left", "Right" is not sufficient (especially in combination). Label individual subplots with letters and reference these in the caption.

---

> ### Author Response · Authors · 2023-11-22
> **Response to Reviewer 3d6u**
>
> We thank the reviewer for their careful reading, and apologize for the lack of clarity. We are clarifying these points in the manuscript, and answer the questions briefly below:
>
> Terms in the Bayesian model are referred to as “predictors”.
>
> The reviewer is correct that the successor representation depends on what policy is encoded in the transition matrix, and that the policy that we use to compute the transition matrix is a random walk policy.
>
> In the simulations in this manuscript, M is computed from the transition matrix (it is not learned). Several previous papers are cited in the manuscript demonstrating that biologically plausible learning rules in a hippocampus-like neural network are able to compute M. For simplicity, in this manuscript, M is computed directly (not through Hebbian plasticity)
>
> We are clarifying the descriptions of the models in the main text, and will post our github repository with all of the code with the archival version of the paper.
>
> In 2.2.2, the true values of the parameters are in the figure titles, and will also be mentioned in the main text. The reviewer is correct that the estimated values match the ground-truth parameters.
>
> A maximum projection is an image that shows, in every pixel, the the brightest value that was achieved throughout the entire movie.
>
> We appreciate the reviewer’s suggestion for different possible model parameterizations.
>
> We will fix the subplot labels.

---

> > ### Comment · Reviewer_3d6u · 2023-11-22
> >
> > While I appreciate the authors' effort, I still believe the current version of the paper is not ready for publication due to lack of clarity and would benefit from further. In addition, I note that the current version of the paper exceeds 9 pages.
> >
> > I have left my evaluation unchanged and encourage the authors to revise, clarify and enrich their paper (e.g. with better explanation of the purpose pf the neural model and the details of the statistical fitting) for future submission at another venue.

---

### Official Review · Reviewer_Vrpj · 2023-10-27

**Soundness:** 3 good
**Presentation:** 4 excellent
**Contribution:** 3 good
**Rating:** 6
**Confidence:** 5

**Summary:**

In this work, the Authors study multi-agent foraging and using various real and simulated data. To this end, they’ve developed a framework that encompasses reinforcement learning, cognitive priors, and statistical description. The Authors tested their framework on synthetic data and then applied it to two datasets on avian foraging: an existing one and a new one that they’ve collected. Their framework has allowed them to confirm previous observations on the existing dataset and to characterize the difference in foraging of various avian species in their new dataset.

**Strengths:**

The goal of the paper is highly important and interesting to the computational neuroscience community. While foraging has been an objective of studies since the 70s, recently its focus has moved from lab experiments to (more) naturalistic behaviors of (groups of) animals interacting in the wild (or almost wild) conditions. In that regard, a computational framework to analyze data from such experiments is timely.

The Authors in this paper aimed to build a unified framework spanning several levels of analysis (i.e., cognitive, network, and algorithmic levels), which, I think, is especially important in the field of foraging because conventionally different levels of analysis were treated separately in the field.

The paper describes a new dataset and lays out the plans to collect more data using an even better technology. This is doubly valuable because new data will allow these and other researchers to test their models of foraging, and also, should additional data be needed to test new hypotheses, the Authors do have a set up framework in place to record such data.

The research is perfectly structured starting with the framework that unifies different levels of analysis, then using synthetic data to validate the framework, later applying the framework to existing data and reproducing the findings in a principled manner, and, finally, introducing new data and obtaining new knowledge off of it.

**Weaknesses:**

I have three small concerns/suggestions. They are intended to suggest some additional analyses, either for the time of the rebuttal (time permitting) or for future research.

First, while the network level of analysis proposed here is indeed biologically plausible, there are no direct links to or mapping onto the brain at this point. An interesting future analysis may involve trying to establish such a mapping (although I do acknowledge that it’s an extremely difficult task if one would like to go beyond high-level similarity). Some prior works have proposed models of how RL models may map onto the brain structure (e.g. see https://www.cell.com/trends/neurosciences/fulltext/S0166-2236(12)00071-9 and https://www.sciencedirect.com/science/article/pii/S0893608002000473 ). It would be interesting to see how the weight matrices proposed in the current work would map onto these previous models and/or onto the brain in general).

Second, the study uses handcrafted features (“proximity”, “trace”) to classify behaviors into handcrafted classes (“random”, “hungry”, “follower”). While both the features and the classes considered here do make sense and are rooted in existing literature, they may mismatch the features and behavioral classes used by the animals and may end up being not expressive enough to describe the observed behavior. A possible way to account for that on the behavioral side is to build a low-dimensional representation of (all available) behavioral readouts and to run any (unsupervised) clustering algorithm on it (e.g. see https://www.cell.com/current-biology/pdf/S0960-9822(17)31604-4.pdf and references therein). On the feature side, a possible strategy is to train an RL model on all available inputs (and/or history thereof), then distill/deduce the relevant (parsimonious) variables (e.g. see https://proceedings.neurips.cc/paper/2020/hash/da97f65bd113e490a5fab20c4a69f586-Abstract.html )

Third, and maybe that’s something I’ve overlooked, but, to my understanding, in the proposed model, as it currently stands, the communication between animals is modeled as a scalar spanning the range from 0 (no communication) to 1 (full reliance on communication). At the same time, animals may transmit signals of different content (e.g. food location vs food quantity); their reliance on the signals of different content may vary (e.g. paying attention to the food location but not to the quantity), and such reliance may be different across species. It looks like this is something that can be analyzed within the existing data and may strengthen the results by discovering the content of the messages passed and received by different species.

While I think that the paper in its current standing is interesting and relevant, updates along the directions outlined above may further expand the scope of the results and strengthen this work.

**Questions:**

Minor comments:

-Page 2. When referencing “Reinforcement learning agents perform well in a variety of foraging-related tasks”, the Mnih et al ATARI paper is perhaps irrelevant, however, https://link.springer.com/article/10.3758/s13415-015-0350-y would offer good support for the cause.

-Would you consider running your model on mouse data? It would be interesting to see how rodent foraging is different from avian foraging. Hopefully, these differences, combined with the overall knowledge of the conditions that the respective species are exposed to, would provide us additional information on the ways the environment shapes foraging strategies.

---

> ### Author Response · Authors · 2023-11-22
> **Response to Reviewer Vrpj**
>
> We thank the reviewer for their insightful suggestions, and their appreciation of the context of this work.
>
> The reviewer pointed out prior studies about mapping from RL models to brain structures. We had originally focused on the correspondence between the hippocampal formation and the successor representation computation – we really appreciate the reviewer’s suggestion to point out correspondences with other brain areas including the basal ganglia.
>
> We appreciate the reviewer’s point that the current set of hand-crafted features will not in general be sufficient to capture animal behavior. One of things we’re most excited about in this approach is the capacity to add flexibly additional potential features. This will be useful for animal behaviors beyond foraging, for example avoiding predators or finding a mate. When additional features that are not in the data are added to the model, the Bayesian inference algorithm should assign zero weight to these features. This was demonstrated for the case of random/hungry/follower birds in Figure 2. For each additional feature, it will make sense to validate that they can be recovered on simulated data.
>
> We agree with the reviewer that animals may transmit and receive signals of different content in different contexts, and that investigating the content of communication would be an interesting extension of our analysis. Unfortunately, our current datasets are insufficient for distinguishing between specific contents of communication, such as between food presence and quantity. However, we are in the process of collecting new datasets, including audio recordings with precise moments that audio communication occurs.
>
> Thank you for pointing out the Constantino and Daw paper. We have included this reference in the main text.
>
> We agree with the reviewer that it would be exciting to try these analyses on mouse data. We intend to publish our Github repository with the archival version of this paper. The repository contains general-purpose functions for fitting these models on new data, and we'd be excited to work with collaborators to tailor the analyses to specific datasets.

---

> > ### Comment · Reviewer_Vrpj · 2023-11-22
> >
> > Thanks for your response.
> >
> > As it seems to me, most of the (other) Reviewers here are uncertain about the (1) state-of-the-art models/results in the field of foraging and, consequently, about the (2) novelty of the proposed results. This is not entirely surprising as these are presumed to be common knowledge within the field of foraging but are not widely known outside of it. Sure, you may also have intended to save some space by skipping over some parts.
> >
> > Here's my suggestion: in the remaining rebuttal time, you may consider writing an extensive overview (as a response to all the Reviewers) covering the state of the field and then listing your contributions on top of that state. I would definitely list a new dataset (which is rare) and analyzing the communication between animals while foraging (which is new). This, I believe, would be the best way forward regarding this discussion.

---

### Official Review · Reviewer_g6r7 · 2023-10-28

**Soundness:** 2 fair
**Presentation:** 2 fair
**Contribution:** 2 fair
**Rating:** 3
**Confidence:** 3

**Summary:**

The paper reports on birds and other social species’ foraging behavior in groups. It models three main behavior types, random motion, following food gradients, and following conspecifics.

**Strengths:**

Refreshing and interesting read about social bird foraging behaviors, their individual differences, and general background.

Appealingly simple models of individual species behavior.

The paper shows that the inference of simple behavioral patterns is possible – at least on the simple level parameters are identifiable.

**Weaknesses:**

The paper does not really have anything novel to offer – or this novelty is at the moment fully hidden. It just reports on how one can generally model multi-agent foraging behavior with very simple techniques. I fully miss any true novelty.

The paper does not really offer any insights. Such as which model may work better to model social foraging behavior etc. There are no ablations etc.

The utility of the presented results is minor. Figure 2 and 3 report performance on simulated data. Parameter identifiability is confirmed, which is good, but also not overly surprising. The actual techniques are reported in the appendix. The results in Figure 3 are summarized but not interpreted. What is the insight of this?

Results in 2.3.1 report on the ability to infer that locusts communicate about food locations (or just observe others consuming food – thus tending to look close-by?). The brevity of the paragraph suggests no interesting insights there.

Results in 2.3.2 report that ducks and small song birds tend to keep other distances to each other, which is also hardly surprising.. and could probably even be measured simply by a summary statistic – a comparison would be necessary to verify that modeling individual foraging decisions is useful here. The tracking technique is also not part of the contribution.

Figure 4 has no 4A 4B 4C 4D to it but is referred in this way.

In general, the evaluations do not sell well. I am left puzzled with what your model now really can show – particularly in relation to other models. Ablation studies or baseline models / standard models in the literature on foraging behavior would need to be offered in comparison to your model to enable judging the quality of your model.

Key techniques used – whether novel or just important to succeed in modeling the foraging behavior – remain fully obscured.

**Questions:**

Note that inverse RL is much older than the work in 2019 that you cite.
In this respect the authors write about information that is inferred from other birds… I think it would be useful not to talk about “amounts” of information but “types” of information (like can they distinguish feeding from just moving around sitting etc… or can they just infer the mere presence of other birds?

Your notation in 2.1.1 only considers model-based RL. I would expect that most birds act habitually in a rather model-free manner – as do your policies I believe?

The neural description then is rather reactive it appears. At least a general appeal to the established fact that brains appear to rather implement generative models (and habitual behavior routines within) would be recommendable.

2.2. How are sensors / strategies / ANN models encoded? At least general intuition about this in the main text would be useful, I think.

---

> ### Author Response · Authors · 2023-11-22
> **Response to Reviewer g6r7**
>
> We appreciate the reviewer’s careful reading and suggestions.
>
> The reviewer’s main concerns were related to whether this work is novel. We believe this work is novel – as mentioned by Reviewer Vrpj, a computational framework to analyze data from real-world experiments is very timely and important for the computational neuroscience community. We are updating the manuscript to make this clearer, and place our results in a fuller context of the existing RL literature.
>
> With respect to parameter identifiability for synthetic datasets, we agree with the reviewer that this is not particularly surprising. These tests were performed as an important methodological validation check before we went on to use the technique on real data.
>
> Related to Figure 3, the results yield both methodological and scientific insights. In terms of methodology, we demonstrate the ability to discriminate between groups of birds which communicate and groups which do not. Bayesian inference methods correctly infer the presence of inter-agent communication from simulated data (Fig. 3 center panels). In terms of scientific insights, communication improves foraging success, as measured as time to first food. Groups which use socially derived information (communicators) achieve greater foraging success than groups who rely solely on self information (non-communicators). This was assessed across different environments. Time to first food reward decreases monotonically with the strength of the communication parameter, across different food patch sizes. Even if one might have had these intuitions to start with, it was important to validate and quantify them with Bayesian probabilistic analysis.
>
> We thank the reviewer for giving us an opportunity to elaborate on section 2.3.1 in which we apply our framework to real data, and we have updated the text.
>
> With respect to the difference between the summary statistics and the Bayesian model (both shown in Figure 4), the summary statistics provide an overview of the correlation between birds’ locations, while the Bayesian model provides predictive information (given a particular sight radius in the model, how predictive are birds’ locations about each other).
>
> With respect to the tracking techniques, while we made use of a pre-existing python package (SLEAP), to our knowledge this was the first application of this package to birds, let alone multi-agent groups of birds in outdoor environments, which required tailoring the data acquisition techniques (thermal + RGB cameras), model parameters, and training, and tailoring multi-agent priors for outdoor data (allowing for individual animals moving in and out of frame).
>
> We agree that validating a model requires comparison of a model with alternative models. The primary contribution of our paper is not the cognitive or neural model itself, but rather the unification of those models with an equivalent statistical model that is amenable to Bayesian inference. To our knowledge, there currently exist no cognitive models of multi-agent foraging susceptible to probabilistic evaluation.
>
> We would like to clarify that the communicator agents share only the locations of food rewards with other agents. However, the flexibility of the modeling approach allows for other types of information to be encoded in the value functions of the agents. We rather present the framework and examples as an invitation to formulate more elaborate and realistic hypotheses in a similar way, and assess alternative hypotheses on real foraging data.
>
> The reviewer raises an interesting question about model-based vs model-free RL. The Successor Representation, which is used in the manuscript, combines features of model-based and model-free RL (https://arxiv.org/abs/1901.11437). The framework we presented could be adapted to more strictly model-free or model-based settings by adjusting the state-space and world-model.
>
> Interestingly, birds are highly intelligent, with behaviors and neural representations that are often compared to primates. Food-caching birds in particular have amazing memories for their environment. They can memorize many different food locations from single exposures, and flexibly use this memory in different contexts, so their behavior does not fit within a strictly model free description.
>
> The reviewer raises an interesting point about generative models implemented by the brain, and we have added reference in the discussion to these ideas.
>
> With respect to sensors and actuators in the agent model, sensors are represented by a sparse encoding of states within an agent’s sight radius, and agents are able to move to adjacent points in the grid world environment – their strategy is to move to the state that is the argmax of the value of all eligible states. This is clarified in the text.

---

> > ### Comment · Reviewer_g6r7 · 2023-11-22
> > **Still lacks both actual insights and proper model comparisons / ablations.**
> >
> > Thank you for your responses.
> >
> > However, I am afraid I do not see my main points addressed. These were: (1) Lack of insights (2) lack of comparison with other models.
> >
> > Answering to your responses paragraph by paragraph:
> >
> > 1. “With respect to parameter identifiability…” -> confirming what I said.
> >
> > 2. “Related to Figure 3…” -> my point was that this is all simulated data, this is all about identifiability – on your own data, generated artificially.
> >
> > 3. On the locust data – there is an apparent error in App.D – which links to Figure 3, which reports the data of your simulated “birds”. You must be referring to Figure 5 here. The insight that locusts communicate with your model seems valid and insightful. I am surprised why you do not integrate this data into the main paper.
> >
> > 4. You emphasize the quality of your tracking techniques. This is good. But this is a conference on “learning representation” not on gathering data. Much more importantly, as I had complained about in my review: Figure 4 is still not properly explained. 4 right side is from modeled data, right? Not from the actual bird data. This feels to me as if you are actively fooling the reader (or confusing me?) – particularly because your write “Ducks” and “Sparrows et al. “ on top of the plots. (by the way, the “et al.” is also very confusing to me).
> >
> > 5. “We agree that validating..” -> it does sound great to have probabilistic evaluation – all models should have this. But, as I said, there is no ablation or comparison to simpler probabilistic, statistical models showing the merit of yours.
> >
> > 6. “ We would like to clarify.. share only the locations of food rewards… ” ->  I don’t see the point. Certainly.
> >
> > 7. “The reviewer raises… model-free vs. model-based…  “ – my point was to evaluate potential alternatives. The relation to a cascade of brain areas that do the proposed computation is extremely far-fetched.
> >
> > 8. “Interestingly, birds are highly intelligent…”  I am well-aware of this fact. That is why I pointed out the rather reactive nature of your policy (one successor step, then winner-takes all action on the believed best reward estimate).
> >
> > 9. Sensor explanations have been clarified to me – thank you.
> >
> > Unfortunately, the paper is now over 9 pages long.
> >
> > Overall, a few things have been clarified by the responses.
> >
> > However, I still do not see the gained insight. Meanwhile, the paper includes three full pages on background. Emphasizing individuality in birds in particular. This was very interesting but was never addressed later. Comparisons to a simpler statistical model of proximities was also not followed up upon. At this point I do not see any true improvements regarding gained insight or a verification of model validity (doing proper model comparison to a baseline model).

---

> > > ### Author Response · Authors · 2023-11-22
> > > **Clarifying data sources -- real data obtained from videos of wild birds**
> > >
> > > The data in Figure 4 is data we collected from real wild birds, foraging outdoors in multi-species groups. The "Ducks" group consisted of Mallards, while the "Sparrows et al" group consisted of White-Throated Sparrows and Tufted Titmice, as is stated in the figure legend. Unfortunately, the reviewer was confused and assumed that this was simulated data.

---

> > > > ### Comment · Reviewer_g6r7 · 2023-11-22
> > > > **plesae then clarify**
> > > >
> > > > your caption Figure 4 reads: "[...] (right) Inferred posterior distriubtions of the proximity coefficient settings of the preferred proximity" -> if this is real wild bird data, where does the "preferred proximity" come from? ... going from 10 up to 80?

---

### Official Review · Reviewer_vuLZ · 2023-10-31

**Soundness:** 2 fair
**Presentation:** 1 poor
**Contribution:** 2 fair
**Rating:** 5
**Confidence:** 2

**Summary:**

This paper proposes computational and implementational descriptions of foraging multi-agent behavior, employing a cognitive model, a neural network model, and a statistical model. The main focus lies on modeling groups of birds in the real world. They simulate behavioral trajectories based on the neural network model and show that proximity and trace can be inferred using Bayesian inference. Finally, the methodology is applied to a locust foraging dataset, revealing differences in proximity preferences among species.

**Strengths:**

The topic of explaining foraging behavior in real-world environments is both intriguing and relevant, contributing to the understanding of animal behavior.

The paper strives to bridge the gap between computational and implementational aspects of behavior, providing a comprehensive view.

The use of a real-world dataset to validate the proposed methodology adds value to the research, demonstrating its practical applicability.

**Weaknesses:**

In general, I found the main point of the paper difficult to grasp and I am still a bit unclear about this. There are different proposals, including different models and their relations, simulated data based on the neural network model, and inferred parameters for the real-world dataset. The experiments as far as I understand only show the relevance of the cognitive model. The implementation as a neural network is used but I would assume that differently designed implementations would yield the same results. So while the implementation may be biologically plausible, it seems to lack concrete evidence to support this type of implementational model if this is what the authors want to claim.

Further, I found section 2.2.2. somewhat unclear in what was done exactly. Do they propose a model and simulate it or do they infer quantities of the model? I did not completely get this from the section but according to the abstract, it seems that Bayesian inference is applied.
For example the sentence "However, some birds also update their expected reward vector at the locations of other birds [...]" could be interpreted in multiple ways, and it's not clear if some birds are modeled differently or if this behavior emerges from uniformly modeled birds. I think the paper would benefit from stating more clearly what was done and what the message of the paper is.

I would also be very careful with formulations such as "When animals decide what action to take, they sample their local environment" or "An animal at state S perceives not only the current state [...]". While these models are often employed in the study of animals, it is unclear whether they represent the actual mechanisms at play in real animals.

**Questions:**

1. What is the primary contribution of the paper? Is it the introduction of different models, the exploration of their interrelations, or the approach to parameter inference? Do the experiments effectively validate this contribution?

2. For the real-world dataset, has the accuracy of your model been measured? It would be beneficial to see a comparison between the simulated behavior and actual data. While inferring different proximity preferences among species is interesting, it doesn't provide insight into how well the model fits the data.

3. When computing posterior distributions of the parameters, the model requires a stochastic component. How is the stochasticity of the agents modeled, particularly when the policy in the model appears deterministic?

---

> ### Author Response · Authors · 2023-11-22
> **Response to Reviewer vuLZ**
>
> We appreciate the reviewer’s feedback on the manuscript. The reviewer’s primary feedback was a lack of clarity – We apologize for this, and are updating the main text to clarify the relevant points.
>
> As mentioned by Reviewer Vrpj, this manuscript addresses a major challenge in computational neuroscience – explaining the behavior of animals in their natural environments. The primary contribution of our paper is to provide a computational framework for reasoning about multi-agent foraging behavior using realistic data. It is very difficult to test cognitive hypotheses, alone because the cognitive variables are quite abstract relative to what can be directly observed. Our framework links the cognitive level with neural and behavioral levels, providing a fairly general framework for researchers to formulate and test hypotheses about the underlying mechanisms of decision-making. For instance, more abstract hypotheses about what the agents value can be tested through this framework at the level of predicting behavior. Since the cognitive model also has an interpretation as a biologically plausible neural network model, it can generate predictions about neural activity in the hippocampus and other brian areas.
>
> The current structure of the paper is as follows: we first introduce a framework that unifies different levels of analysis, then use synthetic data to validate the framework (Figures 2 and 3). We then apply the framework to publicly available data from groups of locusts (Appendix figure) reproducing the findings of the original paper. Finally, we use this analysis framework to distinguish between the behavior of different multi-species groups of wild birds, using video data we recently acquired.
>
> Thank you for identifying section 2.2.2 as a section that needs additional clarity.  We have updated the main text to describe the communication model more explicitly. With respect to the question about updating the reward vector, all birds are modeled uniformly, and reward vectors are updated at the locations of other birds that are at food locations – we apologize for the confusion.
>
> The model has several sources of stochasticity: 1) in the model there is policy noise, and 2) in the inference process there is uncertainty about parameter values due to limited data. In the model, the policy noise comes from the softmax function in the case of communicator birds, and from uniformly random selection of the next location from top n ranked locations in the other simulations.

---

> ### Comment · Reviewer_vuLZ · 2023-11-22
>
> Thank you for your comments.
>
> I am still a bit unsure about the contributions and they are definitely not clearly described in the paper. As I understand it now, you mainly propose the computational model (which you evaluate in your experiments) and give a biologically plausible algorithmic/implementational proposal (which is not evaluated, but used to implement the computational model in your experiments). However, in the "overview" section you start with "We argue that abstract cognitive descriptions of multi-agent foraging behavior can be mapped to a neural network implementation and to a statistical model." This sentence (and the whole section) sounds to me that the computational model would already be well established and your work is to bridge the levels (and for that I would expect experiments to test the algorithmic/implementational hypotheses).
>
> In that whole section, I was not sure of the purpose of what you described there. Some examples:
> "The statistical perspective allows us to infer which of the proposed features best explains the behavior of a particular type of bird in a particular environment." - At what level?
> "To illustrate, we simulated ..." - What exactly did you want to illustrate/show?
> "We investigated whether the benefit of communicating information would be different for different environments, ..." - Why? What does this confirm?
>
> In section 2.2.1. you then claim that the goal is to infer the value from behavior. What is the point of this? Do you additionally want to propose an approach to estimate values and show that it works? Do you want to show the identifiability of the model?
>
> It is not clear to me what the purpose of the model is. If it is to predict birds' behavior, then I wonder why you have not compared your predictions to the actual data. If it is to infer the proximity to other birds, this could have been inferred also with a much simpler model (response of reviewer g6r7).
>
> As for the stochasticity, the value function seems to be computed for a deterministic policy, and this type of policy is even biologically justified in the paper ("Action-selection circuits in the brain are thought to use winner-take-all network dynamics"). Also, in all formulas, the action is defined as maximizing the value function (no softmax). Therefore, I am surprised that you write in your reply that the policy consists of a softmax function. Also, the stochastic transitions should not cause policy noise.

---

### Author Response · Authors · 2023-11-23
**Overview response to reviewers, based on suggestion of Reviewer Vrpj**

At the suggestion of Reviewer Vrpj, we are using the (short) remaining rebuttal time to present an overview response to all the reviewers covering the state of the field of foraging, and describing our contributions on top of that. Reviewer Vrpj said "As it seems to me, most of the (other) Reviewers here are uncertain about the (1) state-of-the-art models/results in the field of foraging and, consequently, about the (2) novelty of the proposed results. This is not entirely surprising as these are presumed to be common knowledge within the field of foraging but are not widely known outside of it."

Understanding real-world animal foraging behavior has been identified as a key open question for understanding intelligent behaviors more broadly, together with other natural animal behaviors. (See, e.g., https://www.nature.com/articles/s41583-018-0010-7; https://link.springer.com/article/10.3758/s13415-018-00682-z; https://pubmed.ncbi.nlm.nih.gov/28182904/; https://pubmed.ncbi.nlm.nih.gov/35609550/; https://www.jneurosci.org/content/41/5/911; https://pubmed.ncbi.nlm.nih.gov/34096743/; https://www.sciencedirect.com/science/article/abs/pii/S2352154622000444).  Several ongoing important lines of inquiry related to foraging are: What learned representations in the brain support animals' ability to learn where to find food? What cognitive computations are they performing? How can we characterize their behavior statistically?

Currently, the study of foraging is siloed -- there has been substantial progress in multiple fields including neuroscience, cognitive science, and statistics (reviewed in Section 1.2 of the manuscript), but it has been difficult to unify/connect results from these different fields. For example, how would foraging strategies would we expect from agents with larger hippocampus, and how would that change the inter-agent movement correlations within a group? Answering this type of question is challenging, because it spans different fields and different levels of abstraction, and to our knowledge, the question was not possible to express in previous frameworks.

The main contributions in this manuscript are as follows:
- Expressing a cognitive model for foraging (Reinforcement Learning using the Successor Representation) in a biologically plausible neural network model. Previously, to our knowledge, only small subsets of the model have been connected to particular brain circuits. In this manuscript, the neuroscience implementation spans the entire decision-making computation from state to action.
- Expressing the same cognitive model in terms of a statistical model, which can be conditioned on data, and where additional terms can be added for each factor that may predict foraging behavior.
- Adding a factor to the model to analyze communication between foraging agents
- Testing the model on simulated data and previously published real data from groups of foraging locusts
- Acquiring new datasets from multi-species groups of birds foraging for food in the wild. This involved collecting the outdoor fieldwork data (IR and RGB videography), as well as customizing the data-processing pipeline to track each bird's trajectory over time.
- Using the model to analyze differences between bird species in the new dataset.

We really appreciate the reviewers' comments and careful reading and suggestions, which we will use to improve the manuscript.

Thanks all!

---

### Meta-Review · Area_Chair_mvfP · 2023-12-05

**Metareview:**

This paper presents a new framework encompassing reinforcement learning, cognitive priors, and statistical description to model multi-agent foraging. The main contribution of the work is in the field of foraging, and only one of the reviewers was an expert in the field. Initially there was some disagreement regarding the suitability of the work for ICLR and the significance of the work. There was unanimous agreement that the paper lacked clarity and should be rewritten for an ICLR audience. During the discussion, the expert reviewer made a compelling case for multiple significant contributions, including (i) a shift from studying individual animals to studying groups of animals; (ii) a novel analysis/model of the interactions between animals, and (iii) a procedure for recording multi-agent foraging data in real environments. During the discussion, all reviewers conceded that these contributions were significant, and there was also unanimous agreement that the paper, as written, was unsuitable for an ICLR audience. It was also noted that the authors could have submitted a significantly revised version of the paper and failed to do so. Overall, the AC agrees with the reviewers that the paper makes interesting contributions, but the paper requires a major rewrite to be suitable for an ML conference.

The AC thus recommends the paper be rejected.

**Justification For Why Not Higher Score:**

As written the paper appears not to be suitable for readers that are not experts in the field of foraging. This paper needs a complete rewrite.

**Justification For Why Not Lower Score:**

NA

---

### Decision · Program_Chairs · 2024-01-16

Reject